# A Comprehensive Review of Clinical Studies Applying Flow-Mediated Dilation

**DOI:** 10.3390/diagnostics14222499

**Published:** 2024-11-08

**Authors:** Yuran Ahn, Nay Aung, Hyo-Suk Ahn

**Affiliations:** 1Division of Cardiology, Department of Internal Medicine, The Catholic University of Korea, Uijeongbu St. Mary’s Hospital, Seoul 06591, Republic of Korea; niceayr@naver.com; 2Catholic Research Institute for Intractable Cardiovascular Disease (CRID), College of Medicine, The Catholic University of Korea, Seoul 06591, Republic of Korea; 3William Harvey Research Institute, Barts and The London School of Medicine and Dentistry, Queen Mary University of London, London E1 2AD, UK; n.aung@qmul.ac.kr; 4National Institute for Health and Care Research Barts Cardiovascular Biomedical Research Centre, Queen Mary University of London, London E1 4NS, UK; 5Barts Heart Centre, St Bartholomew’s Hospital, Barts Health NHS Trust, West Smithfield, London EC1A 7BE, UK

**Keywords:** flow-mediated dilation (FMD), endothelial function

## Abstract

Flow-mediated dilation (FMD) is a noninvasive method to evaluate vascular endothelial function, which manifests the vascular inflammatory response, cell proliferation, and autoregulation. Since FMD is noninvasive and assesses commonly in the brachial artery by ultrasound, compared to other invasive methods such as optical coherence tomography (OCT) and intravascular ultrasound (IVUS), it is widely used to evaluate endothelial function and allows serial assessment. In this review, we present the currently accepted mechanisms and methods of FMD measurement with the studies applied in the current clinical practice using FMD. After all, the association with cardiovascular diseases is of substance, and so we introduce clinical studies of FMD related to cardiovascular disease such as diabetes, hyperlipidemia, chronic kidney disease, coronary artery disease, and peripheral vascular disease. In addition, studies related to pregnancy and COVID-19 were also inspected. Yet, endothelial examination is not endorsed as a cardiovascular prevention measure, for the lack of a clear standardized value methodology. Still, many studies recommend practicable FMD and would be a better prognostic value in the cardiovascular prognosis in future clinical research.

## 1. Introduction

### 1.1. Endothelial Dysfunction and Cardiovascular Disease

Endothelium is a metabolically active organ system that maintains blood vessel homeostasis by blood clotting, inflammation, repairing, and regulating vascular tension, rather than just an anatomical barrier. Nitric oxide (NO), which plays a key role in maintaining the endothelial cells, is a combination of amino acid L -arginine and nitric oxide synthase in endothelial cells. It is generated by endothermic isoform (eNOS). NO interferes with the expression of proinflammatory chemokine and adhesion molecules, limiting the vascular recruitment of white blood cells, blocking the proliferation of vascular smooth muscles and the aggregation of platelets, and preventing the creation of a tissue factor.

Endothelial dysfunction is defined as a reduction in NO levels due to both decreased synthesis and increased consumption, which lead to endothelial impairment. Endothelial cell dysfunction is caused by hypercholesterolemia, hypertension, smoking, diabetes, estrogen deficiency, hyperhomocysteinemia, aging, and other unknown various factors to date (e.g., genetic, and environmental factors). This endothelial cell dysfunction promotes the entire process of atherosclerosis, namely inflammation, oxidation stress, proliferation of smooth muscle cells, platelet activity, and clot formation, eventually causing complications caused by arteriosclerosis.

### 1.2. Evaluation of Endothelial Function Assessment

#### 1.2.1. Venous Occlusion Plethysmography

The earliest tool used was a venous occlusion plethysmography, which was used to assess peripheral endothelial dysfunction. This method needs two cuffs at the arm and wrist to inflate and block intravenous drainage, and simultaneous measurement with cannulation of the brachial artery for vasodilator infusion. This method has been abandoned for its invasiveness and poor reproducibility. Also, it measures the function of conduit vessels, which does not represent the coronary vessels [1].

#### 1.2.2. Quantitative Coronary Angiography and Doppler Flow Wire

Quantitative coronary angiography and intracoronary Doppler flow wire measure coronary diameter changes during coronary angiographic procedures. Combining QCA and Doppler flow wire allows the continuous monitoring of the changes in large coronary arteries diameter and thus assesses both large epicardial coronary artery endothelial function and microvascular endothelial function. The intracoronary infusion of a vasodilator, such as acetylcholine, is used to evaluate endothelial-dependent vasodilation, whereas nitroglycerin is used to assess endothelial-independent vasodilation. Previous studies have shown that in the presence of atherosclerosis, presumably with endothelial dysfunction, vasodilation is insensitive to these vasodilator, resulting in changing the harmony of the vasomotor [2].

#### 1.2.3. Coronary Flow Reserve

Coronary flow reserve (CFR) is the ratio of hyperemic mean blood flow to the resting blood flow. A CFR less than 2 means microvascular dysfunction. There are two ways to evaluate CFR: one is an invasive method that requires a cardiac catheterization, and the other is a noninvasive way, such as positron emission tomography, which measure absolute myocardial blood flow at rest and during hyperemia, myocardial perfusion imaging with single-photon-emission computed tomography (SPECT), which is based on the microsphere method, in which Technetium (Tc)-99m-labeled tracers are taken up by myocardium according to blood flow [3], and echocardiography using Doppler echocardiography measuring coronary flow velocity in the distal portion of the left anterior descending coronary artery (LAD). The vasodilators used for hyperemia are adenosine and dipyridamole [4]. When it comes to the invasive method, it can be measured with a Doppler-tipped coronary guidewire that determines the coronary artery velocity at rest and during hyperemia, usually with intracoronary or intravenous adenosine. As the velocity is proportional to flow, the coronary flow velocity reserve reflects the CFR.

#### 1.2.4. Flow-Mediated Dilation

Owing to its ease of use, reproducibility, and noninvasiveness, FMD is the most generally used method, generally using the brachial artery diameter after stimulation by a vasodilator, such as reactive hyperemia with 5 min sphygmomanometer insufflation at the forearm [5]. Baseline artery diameter and blood flow velocity are measured via duplex ultrasound. Then, the cuff is inflated to about 250 mmHg, which blocks the blood flow to the lower arm for approximately 5 min and induces ischemia to the distal forearm tissue. As the cuff is deflated, the reduction in distal vessels greatly increases the blood flow to the arm. In patients with vascular dysfunction, the degree of dilatation is reduced. The formula of FMD is as follows:$$\frac{\left(maximum\;diameter-baseline\;diameter\right)}{baseline\;diameter\;}\times 100\;\left(\%\right)$$

FMD values in healthy individuals are known as normally distributed, and according to a Japanese analysis, an FMD value above 7.1% is significantly associated with a lower risk of cardiovascular events, and under 2.9% is associated with a higher risk of major cardiovascular event. In this study, the optimal cutoff values of FMD were 7.2% in men and 6.2% in women [6]. Also, there are recent data for normal values of FMD with European individuals suggesting a normal cutoff value of 6.5% as an optimal endothelial function, and the cutoff value for females was lower at 5.7% compared to that of males 6.7% [7]. So far, the exact reason for the difference cutoff value is not known yet, it is supposed that the proportion of risk groups between gender is different.

#### 1.2.5. Pulse Amplitude Tonometry

Arterial pressure pulse waveforms can be quantitatively analyzed, reported noninvasive and reproducible, enable the measurement of pulsatile arterial function, via vasodilator, at rest and after stimulation. Nevertheless, the parameter is subject to many fluctuations. Pulse Amplitude Tonometry uses a fingertip plethysmograph to assess the arterial pulse volume at rest and during hyperemia. The probe is placed on both fingertips of each hand, and the pressure goes up to 10 mmHg below diastolic pressure [8]. After baseline data are acquired, a blood pressure cuff is inflated on one arm to suprasystolic pressures for 5 min. After the cuff release, pulse amplitude increases in the hyperemic finger. Several studies have investigated the relationship between vascular function in the digit and other regions. In patients undergoing coronary angiography, a lower PAT hyperemic response was associated with the presence of coronary endothelial dysfunction measured by acetylcholine response [9]. Also, there are growing cardiovascular risk factors which are linked to impaired digital vascular responses in small studies. Obstructive sleep apnea, preeclampsia, polycystic ovarian syndrome, and mental stress have been associated with a decreased PAT hyperemic response [10,11,12,13].

#### 1.2.6. Thickness of Endothelial Glycocalyx

A layer consisting of proteoglycans and glycoprotein covering the luminal layer of the endothelial cell surface is called the endothelial glycocalyx. As this layer not only preserves endothelial cells but also takes part in regulating vascular hemostasis, injuries to this layer causes endothelial dysfunctions and inflammation. It has been shown that the removal of glycocalyx induces the blocking of NO expression and impairs flow-mediated vasodilation [14,15]. In vivo studies demonstrated that a thin and disrupted glycocalyx layer in regions of vessels presenting a disturbed flow and, in the same areas, atherosclerotic plaques were found [16]. The thickness of the glycocalyx layer ranges from a few nanometer to one micrometer and its measurement is generally dependent on method in vivo or tissue.

### 1.3. The Purpose of This Paper

The aim of this paper was to provide a short overview of the present research and current progress in and importance of endothelial function assessment by using FMD. We present the currently accepted mechanisms and methods of FMD measurement with the studies applied in the current clinical practice using FMD. After all, the association with cardiovascular diseases is of substance, and so we introduce clinical studies of FMD related to cardiovascular disease such as diabetes, hyperlipidemia, chronic kidney disease, coronary artery disease, and peripheral vascular disease.

### 1.4. Materials and Method

Articles were retrieved from PubMed, and the time frame was set to 1 January 2000–31 December 2023. The search was performed by PubMed Advanced Search function within the Title/Abstract fields. The keywords were “Flow mediated dilatation”. The search showed 7734 results, which was too extensive for this review. Consequently, we narrowed these down with “Cardiovascular disease” in combination with each subject such as “Dyslipidemia”, “Diabetes”, “Hypertension”, “Coronary artery disease”, and “Pregnancy”. As we wanted to include the mechanism of FMD, we searched for the keyword “FMD” with “mechanism”, which showed 114 papers that contained recent data. Also, we added the recent outbreak of coronavirus disease (COVID-19) pandemic as well.

## 2. Mechanism of FMD

There are two hypotheses about the induction for FMD: one is a stream of endothelial and smooth muscle hyperpolarization, transport in a retrograde fashion originating at the vasodilated peripheral base towards the interrelated artery, and the other is direct recognition by the endothelial cells of an increased shear stress (Figure 1) [17]. The latter concept brings with another two theories: one is direct communication from the endothelial cell cytoskeleton to the vascular smooth muscle cell, and the other is indirect communication through the glycocalyx [17].

As the distal blood requirement increases, the amount of blood flow increases, and it brings in wall shear stress (blue), which results in arterial dilatation (flow-mediated dilatation (FMD). The presented hypotheses are as follows: (i) a stream of endothelial and smooth muscle hyperpolarization, propagated by endothelial cell hyperpolarizing factor (red); (ii) direct communication via relaxation of smooth muscle (green) by nitric oxide made from the endothelial cells stimulated from dilatation; (iii) indirect communication through glycocalyx, with the cytoskeleton abnormality from the glycocalyx.

### 2.1. Retrograde Spread Theory

There had been a research with a cat femoral artery. The dilator response was obtained when all the nerves to the limb have been cut [18]. Because the conduction velocity was low, and the velocity of neural reflex was not low enough to explain the phenomena, it was reasonable to consider the presence of endothelium-derived hyperpolarizing factor (EDHF), which transmits through junctions between endothelial cell and smooth muscle cells lining in the arterial wall [19,20]. Though we should consider that these findings are made in vitro, they remain highly electrically conductive solutions.

Meanwhile, Kelly and Snow et al. conducted thorough FMD explorations in vivo. They used pig iliac artery more than 3 mm away from the peripheral vessels, and there was an adjustable shunt between artery and vein that allowed discrete, stepwise changes in the diameter and stress of the artery [21]. FMD was observed when the peripheral bed was excluded, in the presence of the shunt. Both NO and electrical transmission via gap junctions are involved in the regulation of the vascular bed. The authors conducted further evaluations to find if both are responsible for dilatation linearly. However, they observed that blocking the production of NO by using the NO synthase inhibitor entirely nullified dilatation [21]. Therefore, it was inconceivable that a hyperpolarizing factor intervenes FMD. Also, further studies were provided against retrograde spread theory [22].

### 2.2. Flow-Mediated Dilation and the Glycocalyx

Both mechanisms that cause an increase in shear stress are directly perceived by the endothelial cells and indirect transmission to the endothelial cytoskeleton, which need mediation by NO. Direct transmission mechanics is activated by the endothelial cytoskeleton alone, and indirect transmission is activated with the other intervention from the endothelial glycocalyx [23,24]. To discover the role of the glycocalyx, Mochizuko et al. and Pahakis et al. showed that disrupting the structure of the glycocalyx also disturbed FMD [25,26]. However, there was an obstruction that their method of disturbing the glycocalyx matrix was causing by using hyaluronidase, which would also have an influence on the intercellular environment, such that it would only support the theory in a limited way. Recent studies from a wide range of fields reveal that organ, tissue, and cell anatomy are each as important for mechanotransduction [27,28,29].

Yao et al. revealed that by using a particular enzyme, heparinase III, to remove the glycocalyx and procreate as if there appeared no flow and confluent endothelial cells respond rapidly to flow by reducing the rate of speed by 40% and increasing the number of vascular endothelial cadherins at cell-to-cell junctions. These responses were not observed in cells mediated with heprinase III [30]. The results were consistent with the theory that the whole system of transmitting from the glycocalyx fibers to the endothelial cell generates flow-induced shear stress on endothelial cells [31,32].

## 3. Assessment of FMD

Flow-mediated vasodilatation with the arm arteries is a widely known method for measuring endothelial function [5]. It is defined by the change in brachial artery diameter using ultrasound. The method was introduced in 1992 [33]. It measures the diameter of arteries reacting to endothelial NO release during reactive hyperemia after a 5 min blood flow blockade by a pressure cuff. The sphygmomanometer cuff is positioned on the forearm distal to the brachial artery where the pressure goes up to 200–300 mmHg [34]. The diameter of brachial artery is measured about 3–5 cm above the ulnar fossa. The technician measures the vessel diameter up to 0.1 mm. The location of the cuff appears to be an important factor for flow dependent dilation (Figure 2). As previously mentioned, the vasodilation is through multiple factors. Putting the cuff at distal to forearm induces NO-dependent dilation, whereas placing it in a position proximal to forearm results in a dilatory response calling multifactorial vasoactive factors. The endothelium-independent dilator response should also be evaluated by sublingual nitroglycerin. This test is necessary to acquire comparable and corresponding dimensions of vasodilation and to control the error of possibility that the endothelium-dependent response may have a chance of being affected by the refitted smooth muscle cell contractility.

### Methodological Issues

Though the method seems simple, some caveats should be considered to produce well-qualified reproducible data. Variations could occur, such as the position of the cuff, the position of the index site of brachial artery, and also the position of the operator. As measured by the naked eye, measuring to an accuracy of 0.1 mm may be at risk of deviation. The relationship between the duration of blockade and its influence on FMD is not clear, but most researchers previously decided to block it for five minutes [35].

## 4. Clinical Studies and Significance of FMD

### 4.1. FMD with Metabolic Syndrome

Metabolic syndrome is a condition including a cluster of risk factors specific for cardiovascular disease, and the factors are abdominal obesity, high blood pressure, impaired fating glucose, high triglyceride levels, and low HDL cholesterol levels [36]. Victoria et al. investigated the relationship between metabolic syndrome and obesity with endothelial function as a prognostic risk factor for CVD (cardiovascular disease). Individuals in the non-obese, non-metabolic syndrome group had a greater FMD than their counterparts. The major finding was that people with metabolic syndrome exhibit endothelial dysfunction, regardless of their obesity status. They showed increased CVD risk in metabolically unhealthy individuals, irrespective of their obesity status, suggesting maintaining good metabolic health may indeed confer a prevention of cardiovascular disease and reduce the risk associated with obesity.

### 4.2. FMD with Hypertension

Chronic hypertension is significantly correlated with vessel inflammation and atherosclerosis [37]. Injured FMD is correlated with hypertension. Carotid IMT (intima-media thickness) sonography and brachial-ankle PWV (pulse wave velocity), collectively, were known to be a useful method to predict the future risk of cardiovascular events in elderly patients. It is well known that male sex and hypertension were common risk factors for vascular complications [38]. There have been many RCT studies of antihypertensive drugs on endothelial function using FMD [39]. Various calcium channel blockers, beta-blockers, angiotensin receptor blockers, and angiotensin-converting enzyme inhibitors were investigated. There were no definite differences between the drugs. Also, there is an interesting report about hypertension and heart structure using FMD. Cetin et al. found an inverse correlation between FMD and left atrial volumes in patients with hypertension [40].

### 4.3. FMD with Diabetes

Reduced endothelial function is common in diabetes patients causing increased incidences of potential severe outcomes including stroke, hypertension, diabetes nephropathy, diabetic foot, and diabetes retinopathy. Therefore, FMD assessment has been demonstrated to predict adverse cardiovascular events [41]. One study showed that an increase in FMD of 1% decreased the risk of a major adverse cardiovascular event by 12% [42]. Lockhart et al. demonstrated that microvascular dysfunction in DM influences blood flow velocity patterns and shear stress [43]. Ying Zhang et al. showed that children who were having type 1 DM had substantially increased IMT and decreased level of FMD [44].

There has been substantial research on the effects of DM medication on FMD, showing improvement in FMD [45,46,47,48,49,50]. Barchetta et al. conducted a study investigating if the elevation of inhibitors of dipeptidyl peptidase 4 (DDP-4) activity are associated with FMD [51]. In a DM group, plasma DDP-4 activity was higher than the control group without DM. Also, DDP-4 activity was associated with a higher BMI and waist circumference and increased levels of plasma transaminases and non-alcoholic fatty liver disease (NAFLD). FMD was decreased and correlated with an elevated plasma DDP-4 activity.

Also, in recent years, the new antidiabetic medication sodium-glucose cotransporter protein 2 inhibitor (SGLT2i) has been associated with reduced CV mortality. In recent times, it has been proven to be effective for heart failure, including preserved ejection fraction, leading to its inclusion in an update to the heart failure guidelines. However, its mechanism of action is not yet fully understood. It has been hypothesized that urinary glucose loss could lead to increased hematocrit, and it might lead to shear stress on endothelial function. In fact, substantial increases have been reported in hematocrit and hemoglobin in a post hoc analysis of the Empagliflozin, Cardiovascular Outcomes, and Mortality in Type 2 Diabetes (EMPA-REG Outcomes) study [52]. Through osmotic diuresis, empagliflozin increases the blood viscosity and subsequently increases shear stress, which may be the major hemodynamic force on endothelial cells, releasing various vasoactive and atheroprotective substances [52]. Concetta et al. showed FMD significantly increased in the empagliflozin group compared to the incretin based group [46]. A similar study was recently published focusing on Dapagliflozin Effectiveness on Vascular Endothelial Function and Glycemic Control (DEFENCE), evaluating the effect of dapagliflozin on endothelial function and glycemic control [53]. A recent meta-analysis conducted to investigate the effects of newer antidiabetic drugs demonstrated that the SGLT2 inhibitor dapagliflozin might influence arterial stiffness [54]. The effects of antidiabetic drugs on FMD are shown in Figure 3. The effect of SGLT-2 inhibitors showed increased FMD, whereas in the group with DPP-4 inhibitors, no specific drug was demonstrated having a significant improvement in FMD. A systematic review of the published research yields rather inconsistent results, based on small randomized clinical trials (Table 1). Still, there is a beneficial effect of SGLT-2 inhibitors on FMD. These benefits could be related to reductions in glucose level, serum uric acid level, and weight, and improvements in oxidative stress and cardiac load. There are many studies on DPP4-inhibitors, but they are more than 10 years old. Large randomized clinical trials shared consistent findings relating to the neutral effect of DPP-4 inhibitors on CVD outcomes. However, there was evidence that saxaglipitin may increase the risk of HF hospitalization [55]. Newer antidiabetic drugs differently affect endothelial function and arterial stiffness, as assessed by FMD and PWV, respectively. Studies were significantly heterogeneous, and therefore, the results should be interpreted with caution. Based on the research so far, we could not conclude that effects of antidiabetic drug are beneficial. More studies are needed to demonstrate the effect of antidiabetic drugs on arterial stiffness.
diagnostics-14-02499-t001_Table 1Table 1Studies of endothelial dysfunction using FMD.AuthorStudy Design/YearAgent/ClassObjectiveAntidiabeticsWidlansky et al. [54]RCT/2016Saxagliptin/DPP-4 inhibitorEvaluate the acute response of the vasculature to DPP-4 inhibition in humans.Shigiyama et al. [56]RCT/2017Linagliptin/DPP-4 inhibitorThe effects of linagliptin on endothelial function.Shigiyama et al. [53]RCT/2017Dapagliflozin/SGLT2-inhibitorDapagliflozin effectiveness on vascular endothelial function and glycemic control.Solini et al. [57]RCT/2017Dapagliflozin/SGLT2-inhibitorWhether dapagliflozin is able to acutely modify systemic and renal vascular function.Kim et al. [58]RCT/2017Vildagliptin/DPP-4 inhibitor To compare the effects of either vildagliptin or glimepiride on glycemic variability, oxidative stress, and endothelial parameters in patients with type 2 diabetes mellitus (T2DM).Kazufumi et al. [49]RCT/2014Sitagliptin/DPP4 inhibitorTo compare the effects of sitagliptin, a DPP-4 inhibitor, and voglibose, an alpha GI, on endothelial function in patients with diabetes.Tatsuya et al. [59]RCT/2016Alogliptin/DPP4 inhibitorWhether dipeptidyl peptidase-4 (DPP-4) inhibition by alogliptin improves coronary flow reserve (CFR) and left ventricular election fraction (LVEF) in patients with type 2 DM and CAD.Ida et al. [60]RCT/2016Trelagliptin/DPP4 inhibitorThe effects of trelagliptin on vascular endothelial function.Dell’Oro et al. [61]RCT/2016Saxagliptin/DPP4 inhibitorThe long-term effects of saxagliptin, as an add-on therapy to metformin, on the above-mentioned variables.Baltzis et al. [62]RCT/2016Linagliptin/DPP4 inhibitorThe effect of linagliptin on surrogates of vascular and mitochondrial function.Ayaori et al. [63]RCT/2013Sitagliptin and Alogliptin/DPP4 inhibitorWhether DPP-4 inhibitors (DPP-4Is) improve endothelial function in T2DM patients.Antihypertensive drugsHeffernan et al. [64]RCT/2011Atonolol vs. Metoprolol succinate/Beta-blocker Atenolol and metoprolol succinate would have disparate effects on vascular function.Takiguchi et al. [65]RCT/2011Olmesartan vs. Amlodipine (ARB vs. CCB)Whether olmesartan and amlodipine would show disparate effects on endothelial function.Yilmaz et al. [66]RCT/2010Amlodipine vs. Valsartan vs. Valsartan/amlodipine (CCB vs. ARB vs. ARB + CCB)Improvement in endothelial function after initiation of angiotensin II receptor blocker (valsartan), calcium channel blocker (amlodipine) therapy, or a combination of both.Jennings et al. [67]RCT/2008Lisinopril vs. Atenolol (ACEi vs. BB)Whether pharmacologic treatment of hypertension normalizes regional cerebral blood flow responses.DyslipidemiaRidker et al. [68]RCT/2017Canakinumab/monoclonal antibody targeting interleukin-1ßWhether canakinumab could prevent recurrent vascular events.Sposito et al. [69]RCT/2022Proteinconvertase subtilisin/kexin 9(PCSK9i)To assess the PCSK9-i effect on the endothelial function of T2D individuals under treatment with SGLT2-i.Antiplatelet agentSchnorbus et al. [70]RCT/2020Clopidogrel vs. prasugrel vs. ticagrelor (ADP receptor inhibitors)The impact of clopidogrel, prasugrel, and ticagrelor on peripheral endothelial function in patients undergoing stenting for an acute coronary syndrome.Lim et al. [71]RCT/2019Ticagrelor vs. clopidogrel (ADP receptor inhibitors)Whether ticagrelor improves endothelial dysfunction in stable patients who survive the acute coronary syndrome.Xu et al. [72]RCT/2021Clopidogrel vs. ticagrelor (ADP receptor inhibitors)To compare the acute effects of ticagrelor versus clopidogrel pretreatment on coronary microvascular function in non–ST-segment–elevation acute coronary syndrome patients.
Figure 3Effect of antidiabetic agents on endothelial dysfunction assessed by FMD. Black dots indicate the mean difference (MD) and the respective 95% confidence intervals in flow-mediated dilatation (FMD) before/after treatment from eligible studies. DPP-4: dipeptidyl peptidase-4; SGLT-2: sodium-glucose cotransporter-2 [49,53,54,57,58,59,60,61,62,63].
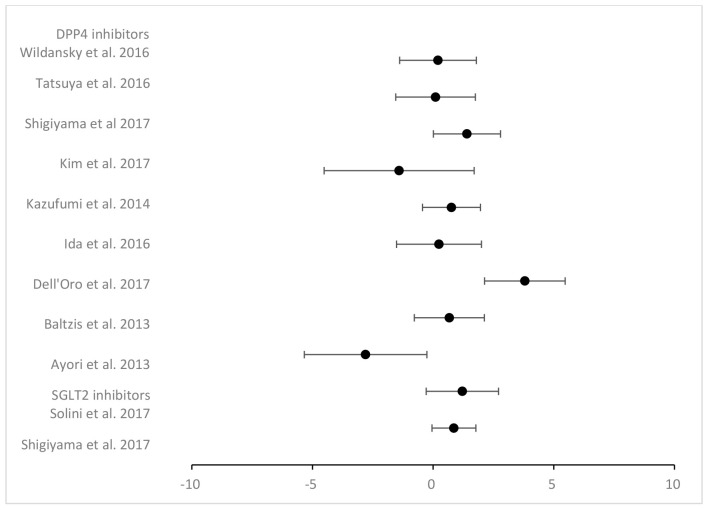



### 4.4. FMD with Dyslipidemia

The mechanisms of the pathogenesis of atherosclerosis are chronic inflammatory process and endothelial dysfunction [34]. Statins are important lipid-lowering drugs, and they additionally have anti-inflammatory abilities. Patients in the dyslipidemia group had lower FMD and higher carotid IMT than the healthy group [68,73]. However, in other studies, Rinkūnienė et al. discovered a significant association between metabolic syndrome, hypertriglyceridemia, and the degeneration of arterial function parameters such as lower IMT as well as higher PWV, but no substantial discrepancy was found in FMD value [74]. In a recent study, Li et al. proved that serum apolipoprotein A-IV levels are associated in flow-mediated dilation in DM patients [75]. They categorized patients into three apoA-IV tertile groups, and FMD was significantly different across three apoA-IV tertiles groups. A recent randomized controlled trial study with evolocumab (proprotein convertase subtilisin/kexin 9, PCSK9-i) was conducted by Sposito et al. [69]. Individuals with type 2 DM were randomized either with empagliflozin only or empagliflozin plus evolocumab. The study has demonstrated that evolocumab intensely reduces LDL cholesterol, non-HDL-C, ApoB, and Lp (a) and reported that evolocumab progressively improves FMD (Table 1).

### 4.5. FMD with Chronic Kidney Disease

It is well known that CVD is the major cause of death in CKD (chronic kidney disease) patients. Patients with CKD have decreased FMD value [76]. Verbeke et al. reported that FMD is significantly decreased in patients with end-stage renal disease (ESRD) without CVD compared to controls and also with coexisting ESRD and CVD. [77]. The study suggested the importance of considering the relationships between shear stress and endothelial function in different clinical conditions. Also, in CKD patients, decreased FMD was significantly associated with proteinuria [78]. In another study, however, Roosa et al. showed that special vascular ultrasound IMT and FMD was not associated with mortality, whereas abdominal aortic calcification score, serum cardiac markers, and echocardiographic parameter E/e’, which usually stands for cardiac diastolic dysfunction, contribute to mortality in CKD patients [79]. The data on the association between endothelial dysfunction and increased mortality in end-stage renal disease are inconsistent and scarce for CKD stage 4–5 patients not on dialysis.

### 4.6. FMD with Cardiovascular Disease

#### 4.6.1. Coronary Artery Disease (CAD)

Due to the increased population of coronary artery disease at a younger age, research efforts on the evaluation and diagnosis of atherosclerotic process and people at risk are important. Gupta et al. [80] showed lower FMD among patients with MI than controls. Many studies observed factors predisposing to coronary heart disease, such as age, male sex, hypertension, diabetes, low FMD, and high IMT [81]. Mangiacapra et al. showed patients with impaired FMD and IMT bring a substantially increased possibility of having CAD in the NINFA study and the addition of FMD and IMT to traditional risk factors improves the discrimination between patients with and without CAD.

#### 4.6.2. Peripheral Artery Disease (PAD)

In another study evaluating peripheral atherosclerosis, beginning with markers of endothelial dysfunction using FMD as indices for clinical outcomes in patients with PAD following PTA (percutaneous transluminal angioplasty) [82], the increase in FMD was higher after PTA in patients with resting pain at baseline compared to those without it.

#### 4.6.3. FMD with Antiplatelet Agent

Regarding antiplatelet studies, various studies demonstrated contrasting results (Table 1). Ramadan et al. suggested that a solitary antiplatelet effect of clopidogrel therapy with stable coronary artery disease not only elicited inflammatory markers but also positive FMD results [83]. Through a systematic analysis of the literature and meta-analyses, Guan et al. showed that ticagrelor, a reversibly binding P2Y_12_ antagonist, has a significant effect on the endothelial function in patients with coronary artery disease [84]. In a recent study, there was a head-to-head comparison of prasugrel and ticagrelor in a randomized controlled trial (RCT). Schnorbus and colleagues showed improved FMD and reduced IL-6 levels after PCI (Percutaneous coronary intervention) in the prasugrel group. They compared the effects of clopidogrel, prasugrel, and ticagrelor on vascular function by assessing the flow-mediated dilation (FMD) of the radial artery, on markers of inflammation by measuring interleukin (IL)-6 and other plasma biomarkers, and on platelet function by analyzing ADP-induced platelet aggregation in NSTE-ACS (non-ST elevation acute coronary syndrome) patients undergoing everolimus-eluting stent implantation [70]. On the other hand, Lim et al. showed that ticagrelor did not improve endothelial dysfunction in stabilized acute coronary syndrome survivors compared to clopidogrel [71]. However, they also pointed out that since endothelial function is greatly influenced by the renin–angiotensin system, the concomitant medications with guideline-directed medical therapy would neutralize the effect of ticagrelor. Also, Mori et al. [85] suggested cilostazol did not affect FMD; however, there was a greater rate of change in the baseline and maximal brachial artery diameters in the Cilostazol group. Patients taking the antiplatelet are highly likely to have other concomitant medication because they usually have other various comorbidities, and it is not usually prescribed alone at clinics, which may explain differences in the findings.

### 4.7. FMD with Pregnancy

There are many studies on the assessment of endothelial function and pregnancy. Throughout gestation, profound hemodynamic change and transformation and body fluid refitting in the body occur. Intravascular blood volume expansion, an increase in cardiac output, and an abrupt reduction in peripheral vascular resistance also occur, being efficient ways to take care of both the mother and fetus. These adaptation developments in shear stress on the vascular wall cause vascular reactivity to self-regulate tone. In contrast, pregnancy encourages hemodilution and it leads to decrease blood viscosity and lower shear forces upon the endothelial wall. Subsequently, local endothelial mechanisms induce a vasomotor response by means of vasodilatation in functionally intact endothelium.

During a healthy pregnancy, endothelium-dependent vasodilatation and FMD increase. Women with a complicated pregnancy had FMD values within the lower range [86]. Lopes van Balen et al. showed that between 15 and 21 weeks of gestation, absolute FMD showed the greatest increase by a mean (95% CI) of 1.89% (0.25–3.53%). This was a relative increase of 22.5% (3.0–42.0%) compared with the reference FMD. In uncomplicated pregnancies, FMD increased in the second and third trimesters. FMD increased progressively in a steady manner but did not reach significance in the first half of the second trimester.

### 4.8. FMD with COVID-19

There is increasing evidence that the angiotensin-converting enzyme 2 receptor (ACE2 receptor) is expressed on endothelial cells (ECs) in the lung, heart, kidney, and intestine, and especially in vessels [87]. With COVID-19 infection, the levels of plasma proinflammatory cytokines (interleukin-1, interleukin-6 (IL-6), and tumor necrosis factor-α), chemokines, von Willebrand factor (vWF) antigen, and factor VIII are elevated. It is reasonable to expect the endothelial dysfunction is associated with COVID-19 with vascular inflammation. Mansiroglu et al. assessed the effects of inflammation on endothelial function in COVID-19 patients with FMD. The study showed that FMD was significantly higher in the control group (9.52 ± 5.98 vs. 12.01 ± 6.18, *p* = 0.01) [88]. Also, Lambadiari et al. found that oxidative stress markers, thrombomodulin, and von Willebrand factor were elevated and FMD values were lower than in controls when measured 4 months after infection. The data suggested a significant association between COVID-19 infection and oxidative stress, endothelial, and vascular dysfunction [89].

## 5. FMD and Future Research

It has been discovered that there is some limitation with FMD methodology about standardization. For example, measurement variance was associated with short response, improper expression of FMD, and poor reproducibility [90]. There was a study published offering an alternative consideration that shear rate–diameter dose–response curves could be another complement to the traditional FMD method and provide a preferable research tool for evaluating CVD risk [91]. Tremblay et al. found that reactive hyperemia-induced flow-mediated dilation does not correspond to the forces that affect the endothelial cell in vivo. Therefore, a modified FMD method was introduced, called sustained stimulus flow-mediated dilation, where the stimuli producing vasodilation are factors such as limb heating, distal vasodilator infusion, and exercise [92]. Currently, reactive hyperemia-induced FMD is widely studied in humans, and further studies are expected to build the clinical utility of sustained stimulus FMD.

## 6. Conclusions

FMD is valuable in the case of patients who have a potential risk of CVD with modifiable cardiovascular risk factors. To review the studies so far, FMD is significantly associated with cardiovascular risk in patients with metabolic syndrome and is reduced compared to controls in patients with not only type 2 diabetes but also type 1 diabetes. Also, there is a positive relationship with patients with hypertension combining with other methods such as carotid IMT sonography and pulse wave velocity. However, the relationship with dyslipidemia with FMD does not show consistent results. For patients with CKD, it is a useful method for evaluating patients with already diagnosed cardiovascular disease.

The most important advantages of the FMD method is that its merits of noninvasive method and relatively low equipment requirements. There are limitations to the method, such as it needing to perform by experienced technicians, and the low reproducibility between different investigators and centers. In addition, quality control should be implemented and care taken to ensure that the test is carried out in appropriate conditions. Also, there are many factors that affect FMD, such as smoking, alcohol, medications, physical activity, stress, outdoor temperature, and seasonality. Therefore, in order to increase the reliability and reproducibility of FMD results, a standardized study protocol in terms of patients and laboratory preparation should be followed when evaluating FMD.

Endothelium is the basis of the cardiovascular system and plays a crucial role in vascular homeostasis. To date, prolific data and research have shown evidence that endothelial dysfunction represents the pathophysiological basis for microvascular coronary artery disease and can advance the CAD, and that endothelial dysfunction is a marker that represents CAD prognosis. Therefore, the assessment of endothelial function by FMD may be an effective and practical tool in making constructive and comprehensive decisions for patients with CVD.

## Figures and Tables

**Figure 1 diagnostics-14-02499-f001:**
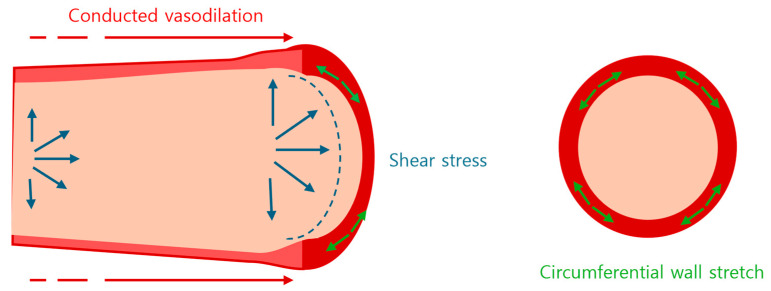
Illustrative diagram of a conduit artery.

**Figure 2 diagnostics-14-02499-f002:**
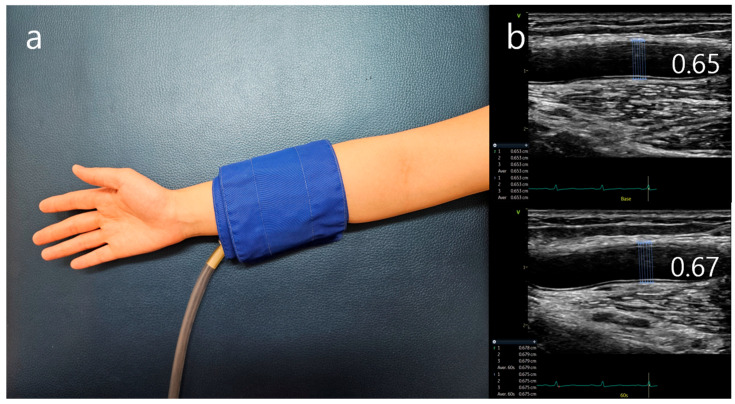
Flow-mediated dilation (FMD). (**a**) Brachial artery diameter is measured at baseline and during hyperemia after inducing 5 min forearm ischemia. Distal location of the cuff. (**b**) FMD is measured, and the diameter of baseline is 0.65 cm and 0.67 cm during hyperemia.

## Data Availability

Data used in this study are available in the medical literature and can be found using the references provided.

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
