# Peer review of "A Comprehensive Review of Clinical Studies Applying Flow-Mediated Dilation"

_diagnostics, 2024, doi:10.3390/diagnostics14222499_

Round 1
Reviewer 1 Report
Comments and Suggestions for Authors
The present study aims to review the current research and progress in endothelial function assessment by using flow-mediated dilation (FMD). Thus, the authors include studies applying FMD in various clinical conditions and highlight the clinical significance of this non-invasive method for evaluating of endothelial function in patients with modifiable risk factors and an increased risk of cardiovascular disease.
Overall, the study is well-written and appropriately structured. However, some points should be addressed.
1. The authors should improve the content of the abstract so as to provide a more comprehensive summary regarding the findings of the studies included in this review.
2. Page 1, line 20: “endothelia” Please correct.
3. All abbreviations throughout the manuscript (CVD, IMT, PWV, PTA, RCT, PCI, NSTE-ACS, etc.) should be explained at the first mention.
4. Section 1.2.3. The authors should mention in brief the assessment of CFR with transthoracic Doppler echocardiography using vasodilators such as dipyridamole and adenosine.
5. Section 1.2.4. A formula for the calculation of FMD as well as reference values for males and females should be given.
6. In the Introduction section, a brief paragraph discussing the measurement of endothelial glycocalyx thickness as a valid method to assess endothelial integrity is highly recommended.
7. The authors should report the factors that affect FMD such as smoking, alcohol, medications, physical activity, stress, outdoor temperature, seasonality. It is essential to emphasize that a standardized study protocol in terms of patients and laboratory preparation should be followed when evaluating FMD in order to increase the reliability of its results (Eur Heart J 2019 Aug 7;40(30):2534-2547. doi: 10.1093/eurheartj/ehz350.)
8. Page 3, line 105-108: Please correct the sentence.
9. Page 6, line 256-257: Please instead of “diabetic drugs” change to “antidiabetic drugs.”
10. Page 6, line 258: “Effect of antidiabetic agents on arterial stiffness assessed by FMD.” Arterial stiffness is assessed by the measurement of pulse wave velocity, not by FMD. A better description and explanation of Figure 3 is also required.
11. Page 9, line 341-342: Please clarify that the numbers are medians with 95% confidence intervals.

Minor editing of English language required.
Author Response
Comments 1: The authors should improve the content of the abstract so as to provide a more comprehensive summary regarding the findings of the studies included in this review.
Response 1: Thank you for your kind recommendations, I agree and therefore added more information of this review paper, Page 1, Abstract, line 20 to 25.
Comments 2: Page 1, line 20: “endothelia” Please correct.
Response 2: Thank you for pointing this out. I corrected "endothelia" to "endothelial"
Comments 3: All abbreviations throughout the manuscript (CVD, IMT, PWV, PTA, RCT, PCI, NSTE-ACS, etc.) should be explained at the first mention.
Response 3: Thank you for pointing this out. I have occordingly modified this abbreviation at the first mention.
Page 6, paragraph 2, line 234
Page 6, paragraph 3, line 243
Page 6, paragraph 3, line 244
Page 8, paragraph 2, line 323
Page 8, paragraph 3, line 342
Page 8, paragraph 4, line 349
Page 9, paragraph 1, line 360
Page 9, paragraph 1, line 362
Page 9, paragraph 1, line 366
Comments 4 : Section 1.2.3. The authors should mention in brief the assessment of CFR with transthoracic Doppler echocardiography using vasodilators such as dipyridamole and adenosine.
Response 4 : we have accordingly modified, added vasodilator briefly. Thank you for pointing this out.
Page 2, paragraph 4, line 77-80
Comments 5 : Section 1.2.4. A formula for the calculation of FMD as well as reference values for males and females should be given.
Response 5 : Thank you, I agree and I revised the contents in Page 3, paragraph 1, line 94-104.
Comments 6 : In the Introduction section, a brief paragraph discussing the measurement of endothelial glycocalyx thickness as a valid method to assess endothelial integrity is highly recommended.
Response 6 : I agree, I've added the contents at Page 3, paragraph 3, line 122 - 131.
Comments 7 : The authors should report the factors that affect FMD such as smoking, alcohol, medications, physical activity, stress, outdoor temperature, seasonality. It is essential to emphasize that a standardized study protocol in terms of patients and laboratory preparation should be followed when evaluating FMD in order to increase the reliability of its results (Eur Heart J 2019 Aug 7;40(30):2534-2547. doi: 10.1093/eurheartj/ehz350.)
Response 7: I totally agreed and added the lines to emphasize the point at Page 10, Paragraph 4, line 439-443.
Comments 8: Page 3, line 105-108: Please correct the sentence.
Response 8 : Thank you for pointing this out. I've seen a lot of error. I've corrected as far as I could.
Comments 9: Page 6, line 256-257: Please instead of “diabetic drugs” change to “antidiabetic drugs.”
Response 9: Thank you for pointing this out. I've corrected them.
Comments 10 : Page 6, line 258: “Effect of antidiabetic agents on arterial stiffness assessed by FMD.” Arterial stiffness is assessed by the measurement of pulse wave velocity, not by FMD. A better description and explanation of Figure 3 is also required.
Response 10: Thank you , I've corrected " arterial stiffness" to " endothelial dysfunction". In page 7, Figure 3. I agree "arterial stiffness" is measured by PWV. I wanted to include that there is still a lack of evidence to conclude whether diabetes drugs have a beneficial effect on FMD.
Comments 11 : Page 9, line 341-342: Please clarify that the numbers are medians with 95% confidence intervals
Response 11 : Thank you for pointing this out. I've corrected the line at page 9, paragraph 3, line 392.
Reviewer 2 Report
Comments and Suggestions for Authors
The manuscript presented is very interesting. The utilization Flow-mediated dilation to evaluate vascular endothelial function is a non-invasive method, easy to be implemented in clinical practice.
However, there are some suggestions for authors:
1. Even if the authors intend to create a narrative review, it is absolutely necessary to respect the structure of a valuable work. Thus, we suggest to its authors the specific objective of the research as well as the methodology by which it was carried out.
2. The authors specify in the title that they want to analyze the studies in the literature that refer to flow mediated dilatation. We suggest the authors to organize a research methodology subchapter in which to specify the duration of the research, the medical databases that were queried, the main and secondary search lines. The studies thus identified should be summarized in a table that includes the authors, the year and the main objective of each study included in the analysis.
3. I recommend the authors to expand the research and include more recent data from the specialized literature (updating the references).
Author Response
Comments 1 : Even if the authors intend to create a narrative review, it is absolutely necessary to respect the structure of a valuable work. Thus, we suggest to its authors the specific objective of the research as well as the methodology by which it was carried out.
Response 1 : Thank you for the kind and clear suggestion. I totally agree and revised at Page 3, paragraph 4, line 136 to Page 4, paragraph 1, line 154.
Comments 2 : The authors specify in the title that they want to analyze the studies in the literature that refer to flow mediated dilatation. We suggest the authors to organize a research methodology subchapter in which to specify the duration of the research, the medical databases that were queried, the main and secondary search lines. The studies thus identified should be summarized in a table that includes the authors, the year and the main objective of each study included in the analysis.
Response 2 : Thank you for pointing this out, I agree with the suggestion. The contents might have a limitation to compass data, and I meant to review the FMD as far as I could, I agree I may have insufficient composition. Therefore I added a table of studies about FMD in this paper at page 8-10.
Comments 3 : I recommend the authors to expand the research and include more recent data from the specialized literature (updating the references).
Response 3 : I agree, thank you for pointing this out. I added more recent data as reference 7, page 3, paragraph 1, line 105. Reference 14, 15, page 3, paragraph 3, line 130.
Round 2
Reviewer 2 Report
Comments and Suggestions for Authors
The authors took into account the recommendations of the reviewers and made improvements to the manuscript, thus making it suitable for publication.
Author Response
I’m really grateful for your help.